# A Study on the Influence of the Use of Sulphur Dioxide, the Distillation System and the Aging Conditions on the Final Sensory Characteristics of Brandy

**DOI:** 10.3390/foods11213540

**Published:** 2022-11-07

**Authors:** María Guerrero-Chanivet, Manuel J. Valcárcel-Muñoz, Dominico Antonio Guillén-Sánchez, Remedios Castro-Mejías, Enrique Durán-Guerrero, Carmen Rodríguez-Dodero, María de Valme García-Moreno

**Affiliations:** 1Departamento de Química Analítica, Facultad de Ciencias, Instituto Investigación Vitivinícola y Agroalimentaria (IVAGRO), Campus Universitario de Puerto Real, Universidad de Cádiz, Puerto Real, 11510 Cádiz, Spain; 2Bodegas Fundador S.L.U. C/San Ildefonso, n 3, Jerez de la Frontera, 11403 Cádiz, Spain

**Keywords:** brandy, wine spirit, aged spirit, sensory evaluation

## Abstract

Brandy is a unique alcoholic beverage obtained from wine distillates. Numerous studies have been published on its physicochemical traits and the effect of certain elaboration variables on them, but not many studies have been carried out from a sensory point of view or that have followed standardized methods applicable to this discipline. This study intends to determine the effect that certain production variables have on the sensory characteristics of brandy. These variables are the following: The use of sulphur dioxide during the fermentation of the base wine, the subsequent distillation system, the alcohol content during aging, the botanical origin of the aging casks, and their toasting degree. For this purpose, the guidelines provided by the ISO standards for sensory analysis have been followed, and chromatic parameters have also been determined. Heavy extractions from *Quercus petraea* casks resulted in brandies with widely varying colors, although these were hard to distinguish using the olfactory and gustatory properties associated with the factors under study. Conversely, those brandies aged in *Quercus alba* casks presented very consistent greenish shades that are not traditionally associated with aged brandy. This lower extraction could explain why the aromatic traits that are found in the fresh spirit are better preserved when this type of oak is used. The spirit obtained through the distillation of SO_2_-free wines aged in *Quercus robur* presented average sensory characteristics: Good color, smooth in the mouth, and medium-intensity oak notes. The distillates that were aged at 55% ABV were later perceived as more aromatically intense with a greater oak note on the palate. On the other hand, the brandies obtained from wines without SO_2_ added were perceived as less alcoholic, sweeter, and more balanced, with a softer oak note.

## 1. Introduction

Brandy is an alcoholic beverage made from wine distillates. According to current regulations [1], it can be made from different types of wine spirits, but a minimum of 50% of the alcohol in the final brandy must come from low- or medium-alcoholic-strength spirits (60–86% ABV—Alcohol By Volume). The distillation method used to produce the wine spirit is a decisive factor regarding its organoleptic characteristics [2]. Two of the most commonly used methods are continuous column distillation [3] and pot still distillation, with the latter able to be carried out in one or two steps [4]. The oenological practices applied to the production of the wine intended for distillation determine its quality [5,6,7] and, consequently, the quality of the spirit that can be obtained from it. The wood type is another key factor that influences the aging process of these spirits [8], since both its botanical origin and the thermal treatment received during the casks’ manufacturing process condition the transfer of the compounds from the barrel to the spirit during the aging stage [9]. *Quercus alba* is the oak species that is most commonly used for the manufacturing of casks for brandy aging. However, other oak species, such as *Quercus petraea* and *Quercus robur* can also be used for this purpose. The casks are usually subjected to a medium toast procedure, although they can also be used after light or heavy toasting. The aging time, the alcoholic strength of the spirit, and the previous use of the casks, including the seasoning of the barrel with sherry wines, are some of the factors that have an effect on the capacity of the spirit to draw compounds from the wood during its aging [10,11,12]. Consequently, brandy composition varies to different degrees, resulting in products that are clearly distinguishable from a physicochemical point of view [13,14,15,16,17].

Some of the works that have been published include data on the effect that a number of factors had on the sensory characteristics of the brandy obtained, but there are few studies that follow the guidelines of sensory analysis standards for brandy, even though sensory analysis has been extensively used for the characterization and quality control of food and beverages [18,19,20,21,22,23,24,25,26]. In the present work, the effect of different processing variables on the sensory characteristics of brandy has been studied. In particular, the following factors have been analyzed: The use of sulphur dioxide during the fermentation of the precursor wine, the distillation system used for the production of the spirit, the alcoholic grade at which the brandy is aged, the toasting degree of the barrels, and the botanical origin of the oak used for the aging process.

## 2. Materials and Methods

### 2.1. Samples

The wines used for different distillations were produced using the Airén grape variety, which has been described by the O.I.V. as the most widely grown white-wine grape variety in the world in the period 2000–2015 [27]. All of the wines came from the winemaking region of Castilla La Mancha, which is the usual origin of the grapes used for the production of the renowned brandy with the European Geographical Indication “Brandy de Jerez” [28].

These wines were obtained from the fermentation of clean must acidified with tartaric acid at a pH of approximately 3.5–3.6 under controlled conditions of temperature between 18 and 25 °C. Some of them were subjected to prefermentative practices involving the addition of sulphur dioxide to the must in the range of 80–120 mg/L prior to the start of the fermentation, while others did not have any sulphur dioxide added to the must at all. The tumultuous and slow fermentations ended one month after the process was started. The wine without sulphur dioxide added was kept cold (10–15 °C) once the fermentation was finished, and after a month, it was decanted and immediately distilled.

The range of spirits included in this work was obtained according to the following processes:WSDD%55: Wine spirit at 70% ABV obtained via double-pot-still distillation (DD) of the freshly fermented wine without the addition of any sulphur dioxide in the process (WS) (the first distillation produced a 30% ABV distillate and the second distillation of this spirit produced a 70% ABV distillate). It was then hydrated to 55% ABV by adding demineralized water before aging.WSD1%55: Wine spirit at 65% ABV obtained via simple distillation (D1) of the wine through a pot still after fermentation and without the addition of any sulphur dioxide (WS). It was then hydrated to 55% ABV using demineralized water before aging.WSD2%55: Wine spirit at 65% ABV obtained using two pot stills that had been set up in series to distill the wine after its fermentation (D2) and without the use of any sulphur dioxide in the process (WS). This system allows the vapors from the first pot still to come into contact with the wine in the second pot still. It was then hydrated to 55% ABV using demineralized water before aging.SCD%55/SCD%65: Wine spirit at 77% ABV obtained via continuous distillation in columns 6 months after the end of fermentation (CD) of selected wine with sulphur dioxide added during the winemaking process (S). It was then hydrated to 55% ABV using demineralized water before aging. Additionally, it was also hydrated to 65% ABV with demineralized water prior to the aging stage.SD2%55/SD2%65: Wine spirit at 65% ABV obtained using two-pot stills configured in series to distill a selected wine 6 months after the end of its fermentation (D2) with the addition of sulphur dioxide (S). It was hydrated to 55% ABV using demineralized water before aging. Additionally, it was also aged at the alcoholic grade of distillation (65% ABV).

New 350 L volume oak casks filled with 335 L of the spirits under study were used for the aging process. The woods used were *Quercus alba* (QA), *Quercus robur* (QR), and *Quercus petraea* (QP), all of which were subjected to two different levels of toasting in the manufacturing of the barrels, light and medium. The length of the aging period was 28 months. Therefore, the brandies produced correspond to VS (very special quality) brandies [29]. Each aging trial was replicated in at least 2 barrels. Each sample evaluated was an equal-volume mixture of the brandies from several barrels of each trial.

All the samples were adjusted to 30% ABV for sensory evaluation. Additionally, in order to reduce the harshness of the spirits and in accordance with European regulations [1], they were smoothed with rectified grape-must concentrate at 840 g/L with up to 4 g/L of inverted sugars. Table 1 shows the data corresponding to the different processing trials that were carried out.

### 2.2. Analytical Methodologies

The analytical characterization of the wines included determining their alcohol content, total acidity, volatile acidity, pH, and sulphur dioxide. Volatile compounds were determined by GC-FID. These analyses were performed on an Agilent 7890B Gas Chromatograph (Agilent Technologies, Santa Clara, CA, USA) coupled to a flame ionization detector. The methods used by the research group in previous works [30] were followed, all of which are the official analysis methods established by the O.I.V. [31].

Regarding the spirits, their alcoholic content (% ABV) was determined by measuring the density of the distillates using a DMA-5000 (Anton-Paar, Ashland, OR, USA). The volatile compounds (total aldehydes, higher alcohols, and esters) were determined by GC-FID, using an Agilent 7890B Gas Chromatograph (Agilent Technologies, Santa Clara, CA, USA) coupled with a Flame Ionization Detector. The method was defined by the researchers in previous work [12]. The CIEL*a*b* space was determined following the standard ISO 11664-4:2008 [32], based on the visible spectra transmittance data of the samples between 380 nm and 830 nm (1 nm resolution), acquired using an Agilent Cary 60 UV-Vis spectrophotometer (Agilent, CA, USA) [33] and glass cuvettes with a 10 mm light path length. The Total Polyphenol Index (TPI) obtained by measuring the absorbance at 280 nm of the conveniently diluted samples was carried out by means of a Perkin Elmer spectrophotometer, Lambda 25 model (Perkin Elmer, Waltham, MA, USA), and 10 mm light-path-length quartz cuvettes. The results were expressed as mg/L of gallic acid equivalent (G.A.E.), for which a calibration curve of gallic acid was plotted in the range of 0–50 mg/L to obtain an R2 = 0.9978. 

All analyses were carried out in triplicate.

### 2.3. Sensory Evaluation Methods

The sensory evaluation sessions were conducted in a tasting room where the influence of external stimuli on the judges could be minimized [34]. The temperature of the room was set at 22 °C. The evaluation of the spirits and brandies was olfactory and gustatory. In all the cases, 35 mL of the sample was served in black blind-tasting glasses labeled with 3-digit random numbers and covered with a lid to favor the concentration of the aromas until the moment of evaluation. The presentation sequence of the samples for the different tests was balanced. Water was used as the rinsing agent, and some breadsticks were also provided as a snack to minimize the effects of alcohol, particularly for those judges who decided not to spit out the samples during the evaluation sessions. The initial tasting panel was made up of 9 selected panelists, who already had some experience in conducting sensory evaluations of different types of oenological samples.

In a preliminary session, the alcoholic strength to be used during the tasting sessions was adjusted by 30 and 36% ABV extended paired-comparison tests [35]. In addition, a triangular test [36] was carried out to evaluate whether any differences could be perceived between brandies aged in medium-toast casks and those aged in light-toast casks, and if so, which were considered to be of better quality.

A quantitative descriptive profile method [37] was used to determine the sensory characteristics of the brandies. The descriptors were selected over 3 sessions according to the guidelines given by standard UNE 87027 [38]. A 9-point interval scale that ranged from 1 for absent and 9 for very intense was used. The tasting panel was trained in the characteristic descriptors of the samples and their subsequent validation, for which 6 sessions were required. Finally, for the double evaluation of the set of samples (26 × 2), 10 sessions were scheduled, at 45 min long each.

### 2.4. Statistical Analysis

For the analysis of variance, correlation matrices, linear discriminant analysis, and cluster analysis of the data, the application Statistica 8.0 (StatSoft, Inc., Tulsa, OK, USA) was used, which also allowed us to construct dispersion graphs. The confidence level of the analysis of variance was 95%. Subsequently, pairwise comparisons were performed by applying the least-significant-difference test. The cluster analyses used the Euclidean distance as the measurement system.

## 3. Results and Discussion

### 3.1. Analytical Characterization of the Wines Used for the Production of the Brandies

Clean musts were obtained from undamaged grapes, which were fermented to produce wines that were suitable for direct consumption. The quality of these wines is crucial, since a deficient wine does not result in a good spirit.

Table 2 shows the basic composition profiles of the four wines that were subjected to distillation in this work. The use of sulphur dioxide should be highlighted as a significant difference between them, which was not added during the prefermentative stage of WSDD%55 and WSD1%55/WSD2%55 wines but was added to SCD%55/SCD%65 and SD2%55/SD2%65. Another difference to take into account is that the unfermented grape juices (musts) that gave rise to WSDD%55 and WSD1%55/WSD2%55 wines exhibited a Baumé degree of approximately 10.5–10.7 Bé, while those musts used to produce SCD%55/SCD%65 and SD2%55/SD2%65 wines were approximately 11.3–11.5 Bé.

At *p*-ANOVA < 0.05, the sulphitation factor appeared to significantly affect the contents of higher alcohols and diethyl succinate, which were lower in wines produced without SO_2_, while some ethyl esters (ethyl decanoate and ethyl dodecanoate) were found in significantly higher concentrations. The quantitative differences between these fermentation products could be partly explained by the yeast-selective effect on the fermentation process that has been widely described for SO_2_ [39]. On the other hand, the time lapse between the end of fermentation and the distillation of the wines that had SO_2_ added could explain the increment in diethyl succinate, as some authors have confirmed through studies on the evolution and loss of freshness of wines during storage [40]. Even though no statistically significant difference was confirmed, the total aldehyde contents (sum of acetaldehyde, acetaldehyde-diethylacetal, and acetoin) was greater in the wines that had been produced using SO_2_. This outcome has been previously described by previous authors who attributed it to a mechanism of resistance to SO_2_ that can be observed in some yeasts [41,42]. Since more than 95% of the total aldehydes corresponded to acetaldehyde, this compound was found in the wines at the concentrations described by other authors [42]. Similarly, pH was somewhat higher in the wines that had been produced without adding any SO_2_. Nevertheless, the four wines presented composition profiles that were consistent with appropriate wines for distillation purposes.

The alcoholic content levels of the wines (10.60–11.50% ABV) were in ranges that had already been described for wines produced using the Airén variety [43,44,45] and were closely related to the Bé levels of their corresponding musts. Airén grapes have a low total acidity, so some acidification of the wines was required to ensure their stability until the moment of their distillation. Values of tartaric acid between 6.41 and 5.19 g/L were reached. Their pH ranged between 3.51 and 3.71, in agreement with the data by Jurado (2016) [45], who highlighted that this variety presented the highest pH and potassium concentrations among the 10 white grape varieties that were part of her study. Volatile acidity levels (0.28–0.51 g/L acetic acid) confirmed no deviations during their alcoholic fermentation [2,45]. SO_2_ contents (36–73 mg/L) remained below the legally established maximum limit of 200 mg/L [46].

Although higher alcohols do not generally have a strong aromatic impact, their excess may mask the aroma of wines [44] or even cause them to be perceived as aggressive, and therefore, in accordance with other authors, total concentrations below 400 mg/L are desirable. This would contribute to their aromatic complexity without being unpleasant [43]. Wines made with SO_2_ presented higher total alcohol contents that were close to this value (340–370 mg/L), while those that did not have SO_2_ added presented lower concentrations (approximately half). Overall, the higher alcohol concentrations of Airén wines included in this study were representative examples of the range described by other authors [46,47,48] and the same is true for their methanol content obtained from grapes’ skin [43,44,48,49]. In turn, 1-hexanol was found at concentrations slightly higher than those described in the literature for this varietal wine. Nevertheless, as it was far from its theoretical perception threshold, there was no risk of it being perceived as undesirable herbaceous notes [43,44,48,50]. On the other hand, 2-phenylethanol, which contributes floral notes, was found in concentrations somewhat lower than those described by other authors for Airén wines, with levels either below or just above its 10 mg/L perception threshold [43,44,48,49,50].

Regarding esters, ethyl acetate (a post-fermentation compound) and fatty acid esters (concentrated in the yeast cells) were generally found at similar contents as those reported in the literature related to Airén wines, while ethyl lactate was actually measured at higher levels than those reported by other authors [2,43,44,48,49,51]. The ratio between aromatic esters and higher alcohols has been proposed as an indicator of the quality of the wines destined for distillation [52]. From this perspective, statistically significant differences were observed between wines made with and without SO_2_, with values ranging from 0.09 (SCD%55/SCD%65) to 0.20 (WSDD%55).

### 3.2. Analytical Characterization of the Fresh Spirits Used for the Production of Brandies

According to European legislation, for spirits to be considered suitable for the production of brandy they must maintain the organoleptic characteristics and volatile components of the raw material from which they come, namely, wine. These volatile substances are defined as the sum of the volatile compounds, excluding ethanol and methanol, of the spirit obtained exclusively by distillation. Table 3 shows the data corresponding to the physicochemical characterization of the distillates obtained from the wines used in this study. The volatile substances represent the sum of volatile acids (expressed as acetic acid), aldehydes expressed as ethanal (i.e., acetaldehyde and the fraction of the latter contained in 1,1-diethoxyethane), higher alcohols (propan-1-ol, butan-1-ol, butan-2-ol, isobutanol, 2-methylbutan-1-ol,3-methyl-butan-1-ol, 1-hexanol, and 2-phenylethanol), and ethyl acetate. Because of their characteristics and the concentrations in which they are found, some of these compounds have been related to notes that contribute to the aroma of the distillates [53]. The amount of volatile substances, or the non-alcohol coefficient of the spirits produced in this work, varied between 169.6 and 370.4 g/HL 100% vol. alcohol, with all values being above the minimum of 125 g/HL 100% vol. alcohol established by the regulations [1].

Furthermore, according to legislation, all the spirits used for the production of the brandies in this research were “wine spirits”, i.e., they were distilled at less than 86% ABV. On the other hand, the Spanish technical standard for the internationally renowned Protected Designation “Brandy de Jerez” classifies the distillation products into three categories according to the resulting alcoholic strength, which is partly related to the distillation system used. Thus, we can obtain low-alcohol wine spirits, traditionally known as “holandas”, with an alcohol content of no more than 70% ABV; medium-alcohol wine spirits, where the alcohol content is between 70 and 86% ABV; and high-alcohol wine spirits, traditionally known as “distilled wine spirits”. In our research, according to their alcoholic strength, those elaborated using pot stills (WSDD, WSD1, WSD2, and SD2) were “holandas”, while those obtained by column distillation (SCD) would be classified as medium-alcoholic-strength spirits.

The column distillate (SCD) presented the highest concentrations of total aldehydes, methanol, higher alcohols (*n*-propanol, isobutanol, *n*-butanol, isoamyl alcohols, and 1-hexanol), and diethyl succinate and the lowest concentrations of higher esters, which is in line with the data reported by other authors [54]. However, this trend is partially echoed by SD2 spirits, which also exhibit high contents of the higher alcohols *n*-propanol, isobutanol, *n*-butanol, isoamyl alcohols, and 1-hexanol, as well as diethyl succinate, while they were low in fatty acid ethyl esters. The application of an analysis of variance at the 95% confidence level for these two spirits versus the other spirits that used SO_2_ during winemaking confirmed a significantly higher content of *n*-propanol, isobutanol, *n*-butanol, isoamyl alcohols, 1-hexanol, and diethyl succinate and a lower level of ethyl esters of hexanoic, octanoic, decanoic, dodecanoic, and tetradecanoic fatty acids, in contrast to what has been described by previous studies on the Airén variety [45]. Some of these differences were already observed in the precursor wines and given that distillation generally involves a concentration of the volatile compounds present in the wine, it should not be surprising that the pattern was repeated in the spirits. The sulfite-free wines were distilled as soon as fermentation was completed, with the presence of yeast residues, so the contribution of fatty acids may have enhanced the fruitiness of their final spirits (WSDD, WSD1, and WSD2), which was in contrast to previous reports [45]. On the other hand, large amounts of higher esters are a source of instability of bottled brandy [55], which should be taken into consideration. The comparison of the brandies obtained by double distillation versus those obtained by column distillation resulted in data that did not match those that had been previously published [54].

In general, total aldehyde contents were lower than previously described for these spirits [45] and below the acceptable limit of 70 g/HL 100% vol. alcohol [47]. As already mentioned, WSDD, WSD1, and WSD2 achieved the lowest values, which seems to be logical given the direct relationship between the use of SO_2_ and the production of acetaldehyde by yeasts during fermentation [41]. Even though the ethyl acetate contents were always below the deviation indicator levels [2,47], its concentrations were somewhat higher than those described in previous studies on distillates of this variety [45] with the lowest values of the ten spirit varieties that were analyzed by the authors. Methanol contents were also somewhat higher than those reported by previous studies and the threshold of 50 g/HL 100% vol. alcohol, which is generally associated with correct pressing [47]. However, it remained well below the maximum permitted limit of 200 g/HL 100% vol. alcohol for all spirits [1]. With respect to the higher alcohols and esters, the concentrations were generally around those described by other authors for Airén spirits [45]. Ethyl lactate was detected at concentrations slightly higher than those previously reported, but still within the range that is considered normal for a pot-still holanda, with no noticeable effect on its sensory characteristics [45]. On the other hand, the concentrations of ethyl succinate in the spirits that used SO_2_ in their production were higher than those described by previous studies [45] and the pattern of previous wines was mirrored.

As a general and most significant conclusion, it can be stated that the levels of the compounds analyzed are within the appropriate quality parameters for spirits intended to be aged in casks.

### 3.3. Color and Phenolic Compounds in Fresh and Aged Spirits

As complementary analytical indicators to the olfactory–gustatory sensory profiles of the aged spirits in this work, the chromatic values and the total polyphenol content were determined. Both of these parameters are related to the extent of extractive and oxidative processes while in contact with the cask wood. Numerous studies have addressed the wood extraction processes that take place during the aging of distillates [54,56,57,58]. With regard to brandy, there has been great interest in polyphenol content, either as a whole or separately [8,13,59], and its relationships with the previous use of the cask [11], aging time [10,60], botanical origin of the oak [8,61], toasting degree of the casks [8,61,62], aging alcoholic degree [11], and final stabilization operations [55,63], but little research has been conducted on the effect of the characteristics of the distillate or the use of sulphur dioxide during the production of precursor wines. According to Table 4, the polyphenol contents of the brandies in this work (70–316 mg/L 100% vol. alcohol) are within the ranges reported by related literature [10,55,61]. Regarding color, a wide variety was registered. The negative values correspond to greenish hues that some of the brandies exhibited, which have been described in previous studies and are associated with a medium toasting degree of the casks’ wood [61], while b* (27–82) and L* (73–95%) values were in good agreement with those reported by different authors [10,61].

A cluster analysis was performed to better visualize the similarities and differences of the samples with regard to all the parameters considered in this study. The result is shown in Figure 1, where it can be observed that the brandies are grouped according to the botanical origin of the oak in which they had been aged.

Clustering of the brandies in contact with *Quercus robur* (QR) is observed, while the two brandies aged in *Quercus petraea* (QP) at 65% ABV can also be found among them. These samples are the closest to the remaining brandies aged in *Quercus petraea* (QP) casks. Those aged in *Quercus alba* (QA) are slightly further apart with the grouping at the greatest distance being reserved for non-aged spirits. By plotting these values on the a*-b* and L*-TPI planes (Figure 2), several conclusions can be drawn.

The relative distances between the brandies in contact with the different oak species that had already been observed in the cluster analysis can be confirmed. Those aged in *Quercus alba* (QA) are characterized by greenish tones (a* negative trait), enhanced brightness (higher L*), and lower total polyphenol contents, which suggests a lower occurrence of the extractive phenomena. The brandies aged in *Quercus robur* (QR) showed reddish tones (a* positive trait) and a greater contribution of yellow hues (greater b*), an intermediate luminosity, and medium polyphenol contents. On the other hand, the brandies in contact with *Quercus petraea* (QP), in contrast to the previous ones, displayed a large dispersion, which could perhaps be attributed to the greater heterogeneity of the casks used for the trials. With intense reddish tones (high a*), they also presented the lowest luminosity (L*) and highest concentrations of total polyphenols, in agreement with the published literature [58]. The results from the analysis of variance confirmed the statistical significance of the aforementioned differences based on the type of oak used (Table 5).

The botanical origin of the oak seems to have a clear effect on the color of the brandy and the amount of phenolic compounds that the distillate extracts from the wood, which is in line with previous data in the literature [33,57]. Perhaps the higher porosity of *Quercus petraea* [64], which facilitates the intake of oxygen and increases the contact area/distillate volume ratio and henceforth the extractive processes, is related to this fact. This intense effect could impair the analysis of the remaining operational factors considered in this work. For this reason, analyses of variance were carried out at the 95% confidence level on different subgroups of samples, which were selected with the objective of determining, preferably, a single effect and its possible interaction with the type of oak. All the parameters were found to be significantly influenced by the various factors analyzed, and interaction with oak origin was confirmed on numerous occasions. This means that the factor in question might affect the brandy differently, depending on the oak with which it was in contact, and it is from this perspective that the results should be analyzed. Table 5 summarizes the results obtained.

In order to determine the influence of the distillation system employed, we compared brandies obtained from their corresponding spirits but adjusted to the same alcoholic strength of 55% ABV using deionized water and aged in medium-toasted casks of the three wood types studied. Having confirmed the interaction, it was necessary to clarify that the spirits aged in *Quercus alba* (QA) showed similar values for L* and a*, regardless of the type of distillate. In contrast, the spirits aged in *Quercus robur* (QR) and *Quercus petraea* (QP), whose distillates had been obtained from double and single distillation in pot stills, respectively, presented the highest color extractions (lower L* and higher a* and b*). According to Delgado et al. (2021) [33], the chemical reactions that contribute to the color of aged spirits critically affect the extraction kinetics of spirits that are rich in non-alcoholic compounds, which could partly explain the differences encountered during the aging of the distillates included in this work. With respect to polyphenol extraction (TPI), only the distillates aged on *Quercus petraea* (QP) exhibited noticeable differences depending on the distillate, being significantly higher in the single-still distillates, followed by the distillates obtained from two serial pot stills and with the use of SO_2_.

The impact of the alcoholic strength of the spirit on its aging was evaluated by comparing two different distillates, one at 65% ABV obtained by serial distillation using two pot stills and the other one at 77% ABV and obtained through column distillation. Each of them was adjusted to two alcoholic grades (55% ABV and 65% ABV) by adding demineralized water before being poured into the casks. That is, the comparison was between SD2%55 and SD2%65 and between SCD%55 and SCD%65. Although the main effects seem to support a higher extraction of color and polyphenols by those distillates aged at 55% ABV, the analysis based on their interactions with the type of oak confirms that only the distillates aged in *Quercus petraea* (QP) presented significantly different chromatic and TPI characteristics between the two alcoholic strengths tested. On the other hand, no significant differences in color and TPI were observed in spirits aged in *Quercus robur* (QR) or *Quercus alba* (QA) that could be associated with the alcoholic strength of the distillate to be aged.

For the evaluation of the effect caused by the usage of sulphur dioxide, two wines differing only in the use of SO_2_ were distilled using two pot stills in series. Subsequently, their alcoholic strength was adjusted to 55% ABV and they were then placed into casks made of the three wood types tested. The aged spirits correspond to WSD2%55 and SD2%55. Regarding color, the interaction with the oak variety means that an analysis must be carried out again for each of the three types of oak. It was concluded from such analyses that no chromatic differences were attributable to the use of SO_2_ in the brandy aged in contact with *Quercus alba* (QA), while some differences could be observed between the brandies aged in contact with *Quercus robur* (QR) and between those aged in *Quercus petraea* (QP) barrels. However, a greater amount of polyphenols was extracted by sulphur-added brandies regardless of the oak wood type, even if the difference in values between QP-aged brandies (270 mg/L G.A.E.) was much more noticeable than between the spirits aged in contact with QR or QA (in which the differences were 43 and 48 mg/L G.A.E., respectively).

### 3.4. Sensory Evaluation of the Samples

#### 3.4.1. Determining the Alcoholic Strength to Be Used at the Tasting Sessions

Given the high alcoholic content of the brandies, the possibility of reducing it in order to facilitate the perception of the differences between the samples and prevent the judges from being affected by sensory fatigue was considered and evaluated. For this purpose, three pairs of coded samples were presented to the judges. The samples were those corresponding to the SCD%55 trials conducted with the three types of oak wood after adjusting their alcoholic grades using deionized water to 36% ABV (usual consumption grade) and 30% ABV (minimum recommended by previous studies [65]). The samples of the two alcoholic grades were presented to the judges the same number of times and in two possible orders of presentation. They were then asked to identify the sample from each pair that they found more difficult to taste. From a total of 31 judgments, 24 confirmed that the 36% ABV samples were perceived as harsher, especially on the palate. This value surpasses the threshold of 21 required by the standard for pairwise comparison tests (to 5% error) [35], therefore the appropriateness of the adjustment to 30% ABV was confirmed. While the biggest differences were found in their mouthfeel where judges rated the 36% ABV samples as less smooth and balanced and more alcoholic and woodier, they also attributed a generally simpler nose character with fewer identifiable notes. It was striking to observe that the descriptive terms that were spontaneously used by the judges coincided, to a large extent, with those considered for the generation of descriptors later on. Based on these results, the sensory study of the brandies was carried out by previously adjusting their alcoholic strength to 30% ABV.

#### 3.4.2. Assessing the Effect of the Toasting Degree of the Casks on the Aged Brandies

Based on a preliminary evaluation and given the large number of samples, it was suggested that light toasting might not be of real interest. In order to verify this hypothesis, olfactory triangular tests were conducted on five types of the tested spirits (WSDD%55, SCD%55, SD2%55, SCD%65, and SD2%65) aged in the three oak varieties (QA, QP, and QR). Each brandy aged in a medium-toast (MT) cask was compared against the same brandy aged in a light-toast (LT) cask. The six possible orders of presentation of the samples (ABB, BAB, BBA, BAA, ABA, and AAB) were balanced to the possible extent.

Of the 15 triangular tests completed, each of which provided nine judgments, only two of them reached six successful judgements. According to the standard [36], this minimum threshold would confirm that the panel perceived a significant difference between brandies depending on the toast level of the cask in these two triangular tests. Although previous physicochemical studies have been able to differentiate the toasting levels [63], some authors have only reported the perception of sensory differences between light and intense toasting [66], in agreement with our results.

An additional question was posed to the judges who were asked to indicate which of the two samples compared in each triad (the singular or the replicate) they would consider to be of higher quality, and to provide reasoning for their choices. Of the six and seven judges who correctly identified the one odd sample in the SCD%65QA and WSDD%55QP triads, five and six of them, respectively, confirmed that they perceived the medium toast oak-aged spirit as more intense, structured, and complex in aroma with well-integrated aging notes while conveying a sense of balance.

This preference for spirits aged in medium-toasted oak confirms the findings reported by previous studies [61]. The sensory study of the samples was therefore narrowed down to brandies aged in contact with medium-toast oak wood.

#### 3.4.3. Selecting the Descriptor

The methodology followed at this stage is in accordance with the standard regarding the identification and selection of descriptors for the generation of a sensory profile [38].

In the first session, the judges were presented with a selection of seven samples from the different processing trials that were representative of the expected extreme values. They were asked to brainstorm as many terms as they found relevant to describe the similarities and differences in the various categories, i.e., odor, basic taste, tactile impression, aroma, and overall sensation. As a result, each judge returned a list of between 15 and 31 descriptors. The panel conductor discarded the hedonic terms (pleasant and unpleasant) and the quantitative ratings (high, medium, and low) from these initial lists, which narrowed the lists down to between 14 and 27 terms, depending on each judge. The lists of terms that had been generated individually in the first session were unified into a single list that was subjected to discussion in the second session in order to confirm that the meaning of each of them was fully understood. A selection of terms describing 26 olfactory notes, 3 basic tastes, 18 tactile impressions, 8 aromas, and 5 global sensations remained (Table 6). In the third session, the brandies from the previous session were evaluated in duplicate with respect to each of the selected descriptors using nine-point interval scales [67].

Different criteria were applied to reduce the number of terms on the list according to the standard. Linear discriminant analyses were also performed to evaluate the multivariate capacity to differentiate sensory profiles according to each processing variable, i.e., distillation method and oak. The descriptors with a relative frequency (% M) greater than 50 were eligible. Furthermore, the descriptors that could be differentiated by an analysis of variance (*p*-ANOVA < 0.2) or those that were part of a multivariate linear discriminant model with a classification percentage greater than 70% were considered. The statistical correlations between terms (analyzed by means of correlation matrices and analysis of clusters of variables) allowed us to discard redundant (highly correlated) and opposite (highly correlated with a negative sign) descriptors.

Finally, the terms selected for the evaluation of the brandies were five olfactory descriptors (aromatic intensity, fruity, vanilla, toasted, and spiced) and five olfactory–gustatory descriptors (sweetness, alcohol, smoothness, oak, and balance). Each is defined in Table 7. The specificity of some of these terms and their alignment with the typicity of brandy are consistent with works on other distillates such as Chilean pisco [68], Cognac [66], South African brandy [69], whiskey [70], rum [71], etc. This contrasts with the generalist terminology used by the O.I.V. to grade the quality of spirits [72], as it could not be otherwise given the variety of products that fall into this category. On the other hand, some of the terms selected by the panel were the same as those used by different authors for sensory studies on aged wine spirits [73,74].

The patterns used for the training sessions were developed using brandies of a long aging period (4–8 years old). Those with a high intensity presented each descriptor with values of 8–9 on a nine-point scale (P8–P9), while those of low intensity, which were differently formulated depending on the descriptor (as indicated in Table 7), were scored approximately 3–4 points (P3–P4). All the patterns were rounded up to 4 g/L of invert sugars using 840 g/L of rectified concentrated grape must.

#### 3.4.4. Tasting Panel Training and Validation

During the following four sessions, the judges’ ability to identify each of the descriptors and their high or low intensities on the nine-point scales was explored and validated. In the first of these sessions, the judges were presented with high-intensity olfactory patterns so that they had a chance to become familiar with them. Then, the identification label on the samples was concealed and they were rearranged for the judges to try and recognize them. The same procedure was followed for the taste patterns. The percentage of successful identifications by the panel ranged from 20% (toasted) to 100% (fruity, spiced, sweetness). In the second session, the judges were presented with the high-intensity patterns used in the previous tasting session and identified by means of labels, so that they could reinforce their learning as an aid for the following test. Then, as a test, each olfactory descriptor’s high- and low-intensity patterns were presented randomly, and the judges were asked to identify the descriptor of the sample pair. In addition, the tasting sheet included the score corresponding to each sample on the nine-point scale (Figure A1). The judges had to first identify the characteristic descriptor of each pair, and then order the samples’ intensities from the lowest to the highest. Once this test was completed, and after a break, the same training procedure was carried out for the gustatory descriptors. A similar third training session was conducted, at the end of which the percentage of successful identification varied between 20% (toasted) and 100% (aromatic intensity, fruity, vanilla, spiced, sweetness, and oak), while the ranking of the intensities reached an average of 84% successful placements. A fourth session was conducted according to the standard [75]. In this session, the judges were presented with two series of six samples (with the necessary break interval between the two series). These samples correspond to a selection of patterns. The scores were treated by two-factor (sample-by-judge) analysis of variance with interaction. The pattern key attributes that were successfully discriminated from the rest of the samples reached 70% (only toasted, aromatic intensity, and balance did not present a *p*-ANOVA lower than 0.05). The interaction between samples and judges scored a *p*-value higher than 0.05 for all the key descriptors, which allowed us to be optimistic about the homogeneity of the panel. However, the judge factor turned out to be significant, and after analyzing the results from each judge, it became apparent that three of them granted different mean scores from the rest of the judges. Therefore, an additional training session on scales was carried out (session 5), followed by a new performance test according to the method used for the first one (session 6). The results improved, as only toasted odor was not successfully discriminated (*p* = 0.453) and the sample-by-judge interaction remained non-significant for all the descriptors (*p* > 0.05). However, the judge factor was again significant (*p* = 0.037), even if just one of the panel members was responsible for these data and it only involved 5 of the 10 descriptors. Therefore, it was decided to remove this judge from the panel. The final number of members in the trained panel was therefore eight (three women and five men, aged from 27 to 60 years).

#### 3.4.5. Sensory Evaluation of the Brandies

Over 10 sessions, 5–6 samples per session were presented to the judges. Table 8 shows the average scores given by the tasting panel to the brandies evaluated. The standard deviations, below 2 points in all the cases, confirmed that the tasting panel’s accuracy was satisfactory. The young spirits were characterized by fruity notes and high aromatic intensities, while the aged brandies, as expected, significantly increased their notes of vanilla, toasted, spiced, and oak.

Given the multiple processing variables considered, in order to at least partially isolate their effects on the sensory characteristics of aged spirits, the panel scores for the different subgroups of samples were subjected to an analysis of variance. For each, the normality and homoscedasticity of the data were previously verified by means of Shapiro–Wilk and Levene tests, respectively, for a confidence of 95%. The *p*-values that confirmed the statistical significance of the effect of a particular factor on a sensory descriptor are indicated by an asterisk. In these cases, the Least Significant Difference (LSD) test was used to compare the different levels. The results are shown in Table 9.

For the analysis of the influence exerted by the distillation system and the type of oak on the sensory profile of the brandies, those obtained by double-pot-still distillation, single distillation, and serial-pot-still distillation (with and without sulphur dioxide addition), as well as continuous column distillation, all of which were adjusted to 55% ABV using deionized water, were compared after aging in medium-toast casks of three oak species (*Quercus robur*, *Quercus petraea,* and *Quercus alba*). A two-factor analysis of variance (distillation-by-oak) with interaction was applied.

The olfactory perception of three of the five descriptors and, most significantly, the fruity one, was affected. The pot still distillation systems without the use of sulphur dioxide (WSDD%55, WSD1%55, and WSD2%55), particularly the double and single distillation systems, resulted in the fruitiest brandies. With close to statistically significant values and p-values just above the 0.05 threshold, these brandies also presented the highest aromatic intensities and the lowest vanilla notes, which could perhaps be attributed to a certain masking effect of the powerful aromatic profile of the young spirit on the aging aromas. Some authors associate a more intense aroma of the distillate obtained by double distillation with the prolonged run times at high temperatures that this process requires and that favors the synthesis of certain volatile compounds [58]. However, this should lead to an enhancement of the toasted notes, which was not observed in our trials. From our point of view, the use of freshly fermented wines and the limited rectification capacity of traditional systems are the main reasons for the high aromaticity of brandies made from WSDD%55, WSD1%55, and WSD2%55 spirits.

The type of oak also affected the fruity note of the brandies, being greater in those aged in *Quercus alba* (QA). The brandies that had been aged in *Quercus robur* (QR) were perceived as less fruity but also had the most intense vanilla notes. On the other hand, the distillates aged in *Quercus petraea* (QP) had more toasted notes, even if the difference with the others was not significant. Other studies [74] have previously investigated the influence of the botanical and geographical origin of oak and other alternative woods on the sensory characteristics of spirits The contribution of the wood also depends on variables that are not always controlled, related to the distillate (pH, alcohol content, and non-volatile components) or the cask itself (porosity, accurate toasting degree, or surface/volume ratio).

The distillation method was also revealed to be rather influential on olfactory and gustatory perceptions, since four of the five descriptors studied showed a significant difference when the distillates were compared. Again, those brandies that had been obtained by pot still distillation and without the use of sulphur dioxide marked the difference, since they were perceived as the smoothest and most balanced in the mouth with the lowest alcoholic sensation. These brandies scored slightly above the average intensity (5 points), while the brandies produced by column distillation (SCD%55) or pot still and SO_2_ (SD2%55) scored at a higher and excessive level, according to the judges’ remarks. Regarding the influence of the type of oak, only the brandies’ smoothness was affected, being significantly poorer in the brandies aged in *Quercus petraea* (QP). At this point, it should be noted that two interactions between distillation and oak were confirmed, which implies that the conclusions required an appropriate revision. Specifically, the differences in smoothness and oak notes of the distillates that had been produced by the traditional method and without sulphur dioxide (WSDD%55, WSD1%55, and WSD2%55) were applicable solely to the brandies aged in *Quercus alba* and *Quercus robur*. As previously mentioned, when analyzing color data, *Quercus petraea* favors an intense extraction of certain compounds, which could have a homogenizing effect on the initial aromatic variability of young eax-de-vie.

Regarding the evaluation of the effect of sulphur dioxide, two wines that differed only in the addition of this component were distilled in two-pot-still systems. Then, their alcoholic content was adjusted to 55% and they were poured into three casks made of the three wood types used in this study. The trials correspond to the aging of WSD2%55 and SD2%55. After applying a two-factor analysis of variance (oak-by-SO_2_) with interactions, no significant interaction could be confirmed for any of the descriptors (*p* > 0.05). This would allow for the analysis on the influence of the use of sulphur dioxide to be simplified. Thus, the brandies that had been produced without the addition of the preservative were perceived in the mouth in a significantly different way for three of the five descriptors (*p* < 0.05); specifically, a less alcoholic and smoother character with a lower intensity of the oak note. In addition, and closely nearing statistical significance, the scores for sweetness and balance were higher. The olfactory perception was characterized by a greater fruity note.

The effect that the alcoholic strength of the young distillate had on the sensory characteristics of brandies was evaluated by conducting trials at 55% ABV and 65% ABV. For this purpose, two of the spirits, the first one obtained at 65% ABV by two-pot-still serial distillation and the second one distilled at 77% ABV using a column, both from wines produced with the addition of SO_2_, were each adjusted to 55% ABV and to 65% ABV using demineralized water before being poured into casks made of the three types of oak wood. The trials corresponded to the aging of SCD%55, SD2%55, SCD%65, and SD2%65. A three-factor analysis of variance (degree of aging, distillation, and oak) with interactions was applied. No interactions between factors were confirmed for any of the descriptors (*p* > 0.05). However, the effect of the distillation method on the spiced note was statistically significant (*p* = 0.001), being higher in brandies that were distilled through two-serial pot stills (intensity of 3.5) compared to those obtained by column distillation (2.7). With respect to the alcoholic degree of aging, those brandies that were aged at 55% ABV were perceived as being more aromatically intense with stronger oak notes on the palate.

Finally, the data were treated as a whole by applying a cluster analysis whose results can be seen in Figure 3. It can be observed that only the young aux-de-vie (non-aged) are clustered apart from the others. The distillates obtained at 65% ABV from pot stills (WSD1%55, WSD2%55, and SD2%55/SD2%65) are closest to each other and are most distant from the double distillates at 70% ABV (WSDD%55). On the other hand, the spirits that had been in contact with wood did not show a clear clustering pattern. The graph displays the double-distilled spirits (WSDD%55) at the ends of the tree, suggesting a certain singularity as was the case with the young spirits. It also shows some groupings of the brandies that shared processing conditions: WSD2%55 (distilled using two serial pot stills without SO_2_ and aged at 55% ABV), SCD%55 (continuous column distillation using SO_2_ and aged at 55% ABV), and SD2%55 and SD2%65 (distilled using two serial pot stills with SO_2_ assistance). In these cases, the type of aging oak did not seem to alter their sensory perception.

## 4. Conclusions

Several studies have been conducted and published on the physical-chemical characteristics of brandy and on the effect of certain processing variables, but few of them focused on their sensory characteristics or implemented the standardized methods established for this discipline.

In terms of olfactory perception, young spirits and brandies obtained by double or single distillation using pot stills and without the addition of SO_2_ were the ones to score highest for fruity notes. The spiced note clearly differentiated the brandies that had been distilled in pot stills from those obtained through column distillation. In the mouth, those without SO_2_ addition and aged in *Quercus robur* or *Quercus alba* were perceived as smoother and richer in subtle oak notes, in contrast to those aged in *Quercus petraea* where the intense extraction seemed to have a homogenizing effect on the initial variety of the young spirits. With respect to the alcoholic grade of aging, the distillates that were aged at 55% ABV were later perceived as more aromatically intense with a greater oak note on the palate. On the other hand, the brandies obtained from wines without SO_2_ added were perceived as less alcoholic with a softer oak note, as well as sweeter and more balanced.

In order to round up this study on the sensory properties of the brandies, the color, defined by its values in the CIEL*a*b* space, and their total polyphenol content have been included as indicators of the extraction of wood components during their aging stage. The lowest values for color and polyphenolic content corresponded to the brandies aged in *Quercus alba* (QA), and except for the effect of sulphur dioxide on their TPI, they were almost unaffected by the other operating factors under consideration, possibly due to the low intensity of the extraction phenomena when this oak species was used. On the other hand, the brandies in contact with *Quercus petraea* (QP) presented chromatic parameters and total polyphenols dependent on the use of sulphur dioxide, their distillation system, and the alcoholic degree of aging, reaching the highest values in general. The brandies aged in *Quercus robur* (QR) casks showed an intermediate chromatic and polyphenolic profile, somewhat closer to that in *Quercus petraea* (QP) and affected to a certain degree by the type of distillate and the addition of sulphur dioxide, but not by the alcoholic degree of aging.

## Figures and Tables

**Figure 1 foods-11-03540-f001:**
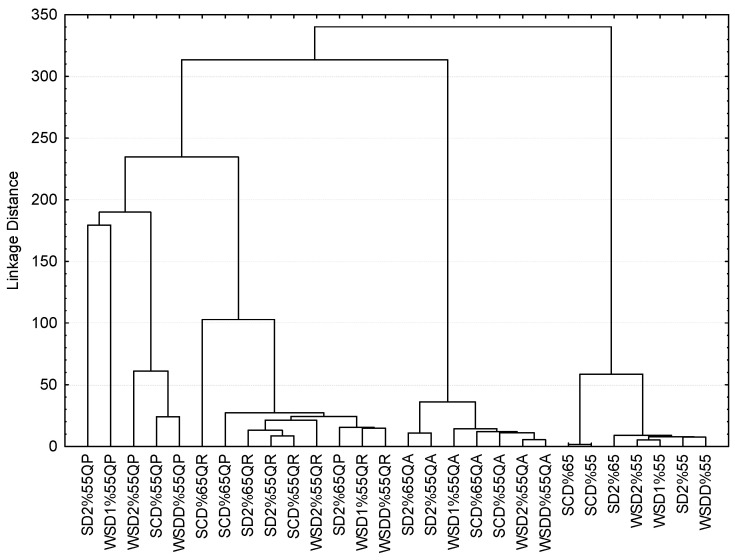
Tree graph representing the cluster analysis of the brandies applied to color and TPI data.

**Figure 2 foods-11-03540-f002:**
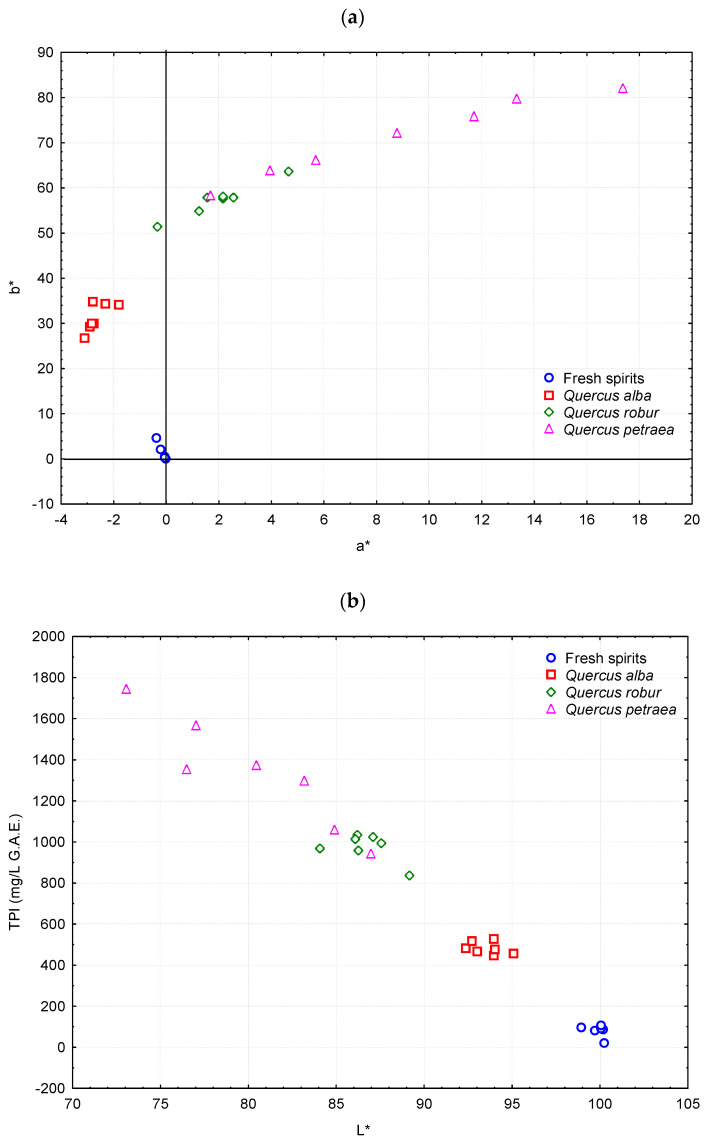
Dispersion plots of the brandies on the a*-b* (**a**) and L*-TPI (**b**) planes.

**Figure 3 foods-11-03540-f003:**
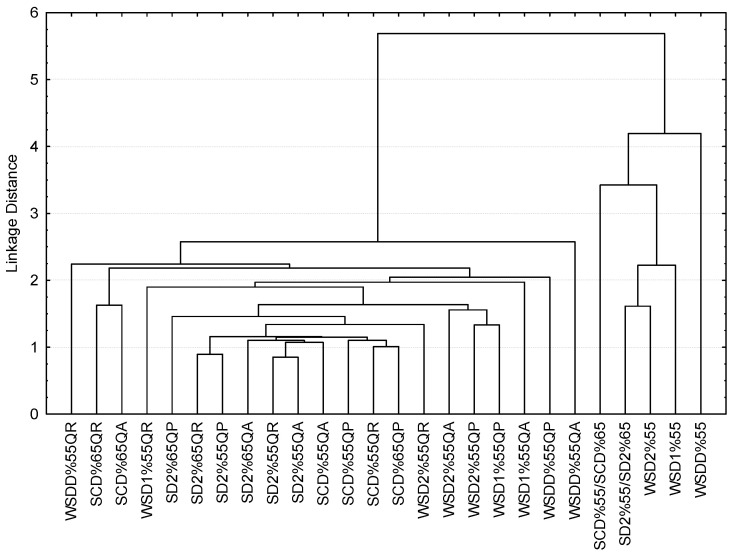
Tree graph representing the cluster analysis of the brandies applied to sensory data.

**Table 1 foods-11-03540-t001:** Brandy-making trials included in this study. Each of these aging trials was conducted in light- and medium-toasted casks.

Sample	Internal Work Code	SO_2_	Distillation System	Distillation Alcoholic Grade	Aging Alcoholic Grade	Oak
WSDD%55QA	AG1_QA	Without (WS)	Double pot still distillation (DD)	70% ABV	55% ABV (%55)	*Quercus alba* (QA)
WSDD%55QP	AG1_QP	Without (WS)	Double pot still distillation (DD)	70% ABV	55% ABV (%55)	*Quercus petraea* (QP)
WSDD%55QR	AG1_QR	Without (WS)	Double pot still distillation (DD)	70% ABV	55% ABV (%55)	*Quercus robur* (QR)
WSD1%55QA	AG2_QA	Without (WS)	Simple pot still distillation (D1)	65% ABV	55% ABV (%55)	*Quercus alba* (QA)
WSD1%55QP	AG2_QP	Without (WS)	Simple pot still distillation (D1)	65% ABV	55% ABV (%55)	*Quercus petraea* (QP)
WSD1%55QR	AG2_QR	Without (WS	Simple pot still distillation (D1)	65% ABV	55% ABV (%55)	*Quercus robur* (QR)
WSD2%55QA	AG3_QA	Without (WS)	Serial distillations with two pot stills (D2)	65% ABV	55% ABV (%55)	*Quercus alba* (QA)
WSD2%55QP	AG3_QP	Without (WS)	Serial distillations with two pot stills (D2)	65% ABV	55% ABV (%55)	*Quercus petraea* (QP)
WSD2%55QR	AG3_QR	Without (WS)	Serial distillations with two pot stills (D2)	65% ABV	55% ABV (%55)	*Quercus robur* (QR)
SCD%55QA	AG4_QA	With (S)	Continuous distillation in column (CD)	77% ABV	55% ABV (%55)	*Quercus alba* (QA)
SCD%55QP	AG4_QP	With (S)	Continuous distillation in column (CD)	77% ABV	55% ABV (%55)	*Quercus petraea* (QP)
SCD%55QR	AG4_QR	With (S)	Continuous distillation in column (CD)	77% ABV	55% ABV (%55)	*Quercus robur* (QR)
SD2%55QA	AG5_QA	With (S)	Serial distillations with two pot stills (D2)	65% ABV	55% ABV (%55)	*Quercus alba* (QA)
SD2%55QP	AG5_QP	With (S)	Serial distillations with two pot stills (D2)	65% ABV	55% ABV (%55)	*Quercus petraea* (QP)
SD2%55QR	AG5_QR	With (S)	Serial distillations with two pot stills (D2)	65% ABV	55% ABV (%55)	*Quercus robur* (QR)
SCD%65QA	AG6_QA	With (S)	Continuous distillation in column (CD)	77% ABV	65% ABV (%65)	*Quercus alba* (QA)
SCD%65QP	AG6_QP	With (S)	Continuous distillation in column (CD)	77% ABV	65% ABV (%65)	*Quercus petraea* (QP)
SCD%65QR	AG6_QR	With (S)	Continuous distillation in column (CD)	77% ABV	65% ABV (%65)	*Quercus robur* (QR)
SD2%65QA	AG7_QA	With (S)	Serial distillations with two pot stills (D2)	65% ABV	65% ABV (%65)	*Quercus alba* (QA)
SD2%65QP	AG7_QP	With (S)	Serial distillations with two pot stills (D2)	65% ABV	65% ABV (%65)	*Quercus petraea* (QP)
SD2%65QR	AG7_QR	With (S)	Serial distillations with two pot stills (D2)	65% ABV	65% ABV (%65)	*Quercus robur* (QR)

**Table 2 foods-11-03540-t002:** Analytical characterization of the wines used for the production of the spirits. A different letter in parentheses indicates a significant-level difference in the sample for *p* < 0.05 (*). The values are expressed as the mean *±* standard deviation.

		WSDD%55 Wine	WSD1%55/WSD2%55 Wine	SCD%55/SCD%65 Wine	SD2%55/SD2%65 Wine	*p*-Anova
Alcoholic content	(% ABV)	10.50 ± 0.06 (a)	10.60 ± 0.08 (a)	11.56 ± 0.07 (b)	11.50 ± 0.05 (b)	0.023 *
Total acidity	g/L Tartaric acid	6.41 ± 0.07	5.33 ± 0.11	5.76 ± 0.12	5.19 ± 0.09	0.312
Volatile acidity	g/L Acetic acid	0.31 ± 0.03 (a)	0.51 ± 0.05 (b)	0.33 ± 0.04 (a)	0.28 ± 0.03 (a)	0.047 *
pH		3.63 ± 0.02 (b,c)	3.71 ± 0.01 (c)	3.56 ± 0.02 (a,b)	3.51 ± 0.02 (a)	0.016 *
Total sulphur dioxide	mg/L	<10	<10	73	36	
Total aldehydes	mg/L	19.3 ± 2.4	23.3 ± 3.5	56.8 ± 2.1	33.9 ± 2.7	0.079
Methanol	mg/L	70.2 ± 1.3 (a)	74.3 ± 2.3 (a)	104.9 ± 1.4 (b)	74.6 ± 2.5 (a)	0.042 *
*N*-Propanol	mg/L	23.9 ± 1.4 (a)	24.6 ± 2.1 (a)	48.3 ± 1.5 (c)	38.3 ± 1.5 (b)	0.002 *
Isobutanol	mg/L	18.1 ± 0.8 (a)	20.5 ± 1.1 (a)	51.2 ± 0.9 (c)	43.1 ± 0.8 (b)	0.011 *
*N*-Butanol	mg/L	1.1 ± 0.1 (a)	1.7 ± 0.1 (a)	2.4 ± 0.1 (b)	2.6 ± 0.1 (b)	0.039 *
Isoamyl alcohols	mg/L	107.3 ± 3.1 (a)	122.2 ± 3.7 (b)	262.4 ± 4.9 (c)	251.3 ± 5.8 (c)	0.005 *
1-Hexanol	mg/L	1.2 ± 0.1 (a)	1.2 ± 0.2 (a)	2.5 ± 0.5 (b)	2.9 ± 0.5 (b)	0.041 *
2-Phenylethanol	mg/L	8.8 ± 1.7 (b,c)	5.1 ± 1.1 (a)	7.8 ± 0.8 (a,b)	11.3 ± 0.8 (c)	0.046 *
Ethyl acetate	mg/L	38.1 ± 2.5 (a)	36.9 ± 3.0 (a)	38.4 ± 2.0 (a)	49.1 ± 1.7 (b)	0.044 *
Ethyl lactate	mg/L	18.6 ± 2.9 (a)	40.6 ± 3.9 (c)	30.1 ± 2.6 (b)	63.4 ± 1.3 (d)	0.003 *
Diethyl succinate	mg/L	0.1 ± 0.1 (a)	0.1 ± 0.1 (a)	1.3 ± 0.2 (b)	1.2 ± 0.3 (b)	0.025 *
Ethyl hexanoate	mg/L	0.4 ± 0.1	0.6 ± 0.2	0.6 ± 0.2	0.3 ± 0.1	0.723
Ethyl octanoate	mg/L	1.4 ± 0.2	1.6 ± 0.2	1.6 ± 0.2	1.5 ± 0.2	0.881
Ethyl decanoate	mg/L	2.8 ± 0.3 (b)	2.4 ± 0.3 (b)	1.2 ± 0.1 (a)	0.9 ± 0.1 (a)	0.037 *
Ethyl dodecanoate	mg/L	2.1 ± 0.3 (b)	2.3 ± 0.4 (b)	0.4 ± 0.1 (a)	0.6 ± 0.1 (a)	0.031 *
Ethyl tetradecanoate	mg/L	0.1 ± 0.0	0.1 ± 0.0	0.1 ± 0.0	0.1 ± 0.0	0.933
Ethyl hexadecanoate	mg/L	0.3 ± 0.1	0.2 ± 0.1	0.3 ± 0.1	0.4 ± 0.1	0.805

**Table 3 foods-11-03540-t003:** Analytical characterization of the young spirits obtained. ND: Non detected. A different letter in parentheses indicates a significant-level difference in the sample for *p* < 0.05 (*). The values are expressed as the mean ± standard deviation.

		WSDD	WSD1	WSD2	SCD	SD2	*p*-Anova
Alcoholic content	(% ABV)	70.0 ± 0.2	65.0 ± 0.1	65.0 ± 0.0	77.0 ± 0.2	65.0 ± 0.1	
Total aldehydes	mg/L	70.8 ± 2.8 (a)	91.4 ± 3.9 (b)	66.7 ± 2.8 (a)	275.0 ± 11.3 (c)	121.7 ± 5.9 (b)	0.027 *
Ehyl acetate	mg/L	153.1 ± 3.6 (a)	212.5 ± 5.1 (c)	174.6 ± 4.7 (b)	171.8 ± 4.0 (b)	247.9 ± 5.8 (d)	0.011 *
Methanol	mg/L	415.4 ± 8.0 (a)	414.1 ± 7.9 (a)	415.6 ± 8.0 (a)	636.0 ± 12.3 (b)	430.8 ± 8.3 (a)	0.022 *
*N*-Propanol	mg/L	150.5 ± 3.2 (b)	149.5 ± 3.7 (b)	135.7 ± 2.9 (a)	317.1 ± 6.8 (d)	211.5 ± 4.5 (c)	0.014 *
Isobutanol	mg/L	116.6 ± 2.2 (a,b)	124.8 ± 3.9 (b)	106.5 ± 3.1 (a)	334.7 ± 9.1 (d)	236.9 ± 6.4 (c)	0.007 *
*N*-Butanol	mg/L	ND	ND	ND	14.9 ± 0.5	14.0 ± 0.5	0.477
Isoamyl alcohols	mg/L	696.4 ± 9.9 (b)	740.8 ± 10.5 (c )	645.5 ± 9.1 (a)	1738.3 ± 24.8 (e)	1372.0 ± 19.5 (d)	0.002 *
1-Hexanol	mg/L	7.0 ± 0.2 (a)	6.9 ± 0.2 (a)	6.7 ± 0.2 (a)	22.3 ± 0.6 (c)	15.5 ± 0.4 (b)	0.013 *
2-Phenylethanol	mg/L	4.8 ± 0.2 (a,b)	5.9 ± 0.2 (b)	6.3 ± 0.3 (b)	4.2 ± 0.2 (a)	13.7 ± 0.5 (c)	0.029 *
Ethyl lactate	mg/L	63.1 ± 2.0 (a)	88.4 ± 2.9 (b)	99.1 ± 3.1 (b,c)	111.5 ± 3.7 (c)	175.0 ± 5.5 (d)	0.033 *
Diethyl succinate	mg/L	0.4 ± 0.1 (a)	0.2 ± 0.1 (a)	0.2 ± 0.1 (a)	10.3 ± 0.3 (c)	7.6 ± 0.3 (b)	0.002 *
Ethyl hexanoate	mg/L	5.4 ± 0.2 (c)	5.8 ± 0.5 (c)	4.5 ± 0.3 (b)	3.6 ± 0.2 (a)	3.5 ± 0.2 (a)	0.040 *
Ethyl octanoate	mg/L	14.6 ± 0.3 (c)	13.7 ± 0.3 (c)	13.2 ± 0.3 (c)	11.7 ± 0.3 (b)	7.8 ± 0.2 (a)	0.042 *
Ethyl decanoate	mg/L	25.8 ± 0.6 (d)	25.4 ± 0.8 (d)	17.9 ± 0.5 (c)	8.4 ± 0.3 (b)	5.9 ± 0.2 (a)	0.021 *
Ethyl dodecanoate	mg/L	18.3 ± 0.5 (c,d)	21.0 ± 0.8 (d)	15.8 ± 0.5 (c)	1.8 ± 0.3 (a)	2.9 ± 0.2 (b)	0.036 *
Ethyl tetradecanoate	mg/L	1.7 ± 0.1 (b)	2.7 ± 0.1 (c)	2.8 ± 0.1 (c)	0.5 ± 0.2 (a)	0.7 ± 0.2 (a)	0.005 *
Ethyl hexadecanoate	mg/L	4.8 ± 0.2 (b)	6.1 ± 0.2 (c)	7.3 ± 0.2 (d)	2.4 ± 0.2 (a)	3.7 ± 0.2 (a)	0.049 *

**Table 4 foods-11-03540-t004:** Chromatic characteristics and TPI of the young and aged spirits. The values are expressed as the mean *±* standard deviation.

	L*	a*	b*	TPI (mg/L G.A.E.)
WSDD%55	99.0 ± 0.5	−0.3 ± 0.1	4.5 ± 1.6	95.6 ± 12.5
WSDD%55QA	92.4 ± 0.1	−1.8 ± 0.1	34.1 ± 1.0	478.1 ± 16.9
WSDD%55QR	84.1 ± 2.3	4.7 ± 2.5	63.6 ± 4.8	967.8 ± 110.2
WSDD%55QP	76.5 ± 3.3	13.3 ± 3.6	79.7 ± 4.6	1352.1 ± 182
WSD1%55	99.7 ± 0.1	−0.2 ± 0.0	1.9 ± 0.0	77.1 ± 1.1
WSD1%55QA	94.0 ± 0.0	−2.9 ± 0.0	29.3 ± 0.0	442.2 ± 1.3
WSD1%55QR	86.3 ± 0.0	2.2 ± 0.0	57.5 ± 0.0	954.7 ± 5.4
WSD1%55QP	73.1 ± 0.0	17.4 ± 0.0	82.0 ± 0.0	1743.2 ± 2.8
WSD2%55	100.2 ± 0.0	−0.0 ± 0.0	0.3 ± 0.0	81.9 ± 0.5
WSD2%55QA	94.1 ± 0.0	−2.7 ± 0.1	30.0 ± 0.0	475.0 ± 0.8
WSD2%55QR	87.6 ± 0.1	1.3 ± 0.1	54.9 ± 0.1	990.0 ± 0.5
WSD2%55QP	83.2 ± 0.1	5.7 ± 0.0	66.2 ± 0.2	1293.6 ± 1.02
SCD%55/SCD%65	100.2 ± 0.1	0.0 ± 0.0	0.0 ± 0.0	17.1 ± 0.0
SCD%55QA	93.1 ± 0.0	−2.8 ± 0.0	34.8 ± 0.0	465.0 ± 0.0
SCD%55QR	87.1 ± 0.0	1.6 ± 0.0	57.7 ± 0.0	1024.0 ± 1.6
SCD%55QP	80.5 ± 0.0	8.8 ± 0.0	71.9 ± 0.0	1374.1 ± 0.84
SCD%65QA	95.1 ± 0.0	−3.1 ± 0.0	26.8 ± 0.0	456.3 ± 5.0
SCD%65QR	89.2 ± 0.0	−0.3 ± 0.0	51.3 ± 0.0	836.7 ± 3.6
SCD%65QP	85.0 ± 0.0	4.0 ± 0.0	63.9 ± 0.0	1058.9 ± 0.4
SD2%55/SD2%65	100.1 ± 0.1	−0.0 ± 0.0	0.1 ± 0.0	103.4 ± 2.7
SD2%55QA	94.0 ± 0.0	−2.8 ± 0.0	30.0 ± 0.0	523.9 ± 2.9
SD2%55QR	86.2 ± 0.0	2.2 ± 0.0	58.0 ± 0.0	1032.4 ± 3.6
SD2%55QP	77.0 ± 0.0	11.7 ± 0.0	75.7 ± 0.1	1564.0 ± 2.69
SD2%65QA	92.7 ± 0.0	−2.3 ± 0.0	34.3 ± 0.0	514.0 ± 1.7
SD2%65QR	86.1 ± 2.3	2.6 ± 2.2	57.7 ± 6.3	1011.0 ± 149.5
SD2%65QP	87.0 ± 0.0	1.7 ± 0.0	58.3 ± 0.0	939.3 ± 197.5

**Table 5 foods-11-03540-t005:** Results from the analysis of variance applied to the data on color and total polyphenol content (TPI) of the brandies. A different letter indicates a significant level difference of the factor for *p <* 0.05 (*). The values are expressed as the mean ± standard deviation.

		L*	a*	b*	TPI (mg/L GAE)
**Factor: Oak**	*p-*Oak	0.000 *	0.000 *	0.000 *	0.000 *
	*Quercus alba*	93.5 ± 0.7 (c)	−2.6 ± 0.4 (a)	31.6 ± 2.5 (a)	476.8 ± 28.7 (a)
	*Quercus robur*	86.3 ± 1.5 (b)	2.4 ± 1.5 (b)	58.3 ± 3.4 (b)	993.8 ± 48.8 (b)
	*Quercus petraea*	77.1 ± 3.9 (a)	12.5 ± 4.5 (c )	76.7 ± 6.2 (c)	1464.8 ± 185.1 (c)
**Factor: Distillation system**	*p-*Distillation	0.000 *	0.000 *	0.000 *	0.002 *
*p-*Distillation x Oak		0.000 *	0.000 *	0.003 *	0.000 *
Double pot still distillation (WSDD%55)		84.3 ± 7.3 (a)	5.4 ± 7.1 (b)	59.1 ± 20.9 (d)	932.7 ± 403.3 (a)
Simple pot still distillation (WSD1%55)		84.5 ± 9.5 (a)	5.6 ± 9.4 (b)	56.3 ± 23.6 (c)	1046.7 ± 586.2 (b)
Serial distillations with two pot stills. Without SO_2_ (WSD2%55)		89.3 ± 4.7 (c)	0.6 ± 3.5 (a)	47.2 ± 16.4 (a)	919.3 ± 369.8 (a)
Continuous distillation in column (SCD%55)		88.2 ± 5.2 (b,c)	1.3 ± 4.7 (a)	51.4 ± 16.2 (b)	954.2 ± 409.9 (a)
Serial distillations with two pot stills. SO_2_ added (SD2%55)		87.5 ± 7.0 (b)	2.1 ± 5.9 (a)	50.3 ± 20.0 (b)	1039.5 ± 464.3 (b)
**Factor: Aging alcoholic grade**	*p-Alc. grade*	0.000 *	0.000 *	0.001 *	0.000 *
*p-*Alc. grade x Oak		0.001 *	0.000 *	0.033 *	0.000 *
55% ABV (SCD%55, SD2%55)		87.8 ± 5.8 (a)	1.7 ± 5.1 (b)	50.8 ± 17.2 (b)	996.8 ± 419.9 (b)
65% ABV (SCD%65, SD2%65)		89.2 ± 3.9 (b)	0.4 ± 2.8 (a)	48.7 ± 14.3 (a)	802.7 ± 256.9 (a)
**Factor: SO_2_ added in musts**	*p*-SO_2_	0.000 *	0.000 *	0.000 *	0.000 *
	*p*-SO_2_ x Oak	0.000 *	0.000 *	0.000 *	0.000 *
Without (WSD2%55)		89.3 ± 4.7 (b)	0.6 ± 3.5 (a)	47.2 ± 16.4 (a)	919.3 ± 369.8 (a)
With (SD2%55)		87.5 ± 7.0 (a)	2.1 ± 5.9 (b)	50.3 ± 20.0 (b)	1039.5 ± 464.3 (b)

**Table 6 foods-11-03540-t006:** List of descriptors resulting from the brainstorming phase and later refined by the panel leader. This list was used as the first step toward the selection of the appropriate descriptors according to the standard.

Olfactory Notes	Tastes	Tactile Sensations	Aromas	Overall Sensations
Alcoholic	Sweetness	Padded	Caramel	Complexity
Aniseed	Acidity	Alcohol	Oxidative sweetness	Balanced
Aromatic intensity	Bitterness	Burning	Spiced	No edges
Caramel		Astringency	Nuts	Full
Clove		Velvety	Herbaceous	Persistence
Coconut		Hot	Noble wood	
Coffee		Fleshy	Oak	
Floral		Consistency, body	Vinous	
Fruity		Estructured		
Glue		Fluid		
Herbaceous		Alcohol integration		
New Wood		Good throat pass		
Noble wood		Pungent		
Nuts		Rough		
Oak		Drying		
Raisins, dried fruits		Smooth		
Spiced		Tannic		
Sweet		Unctuous		
Toasted				
Toffee				
Tropical fruits (pinepple, banana,...)				
Vanilla				
Varnish, solvent				
Vinous				
White fruits (apple, pear,...)				
Wine lees				

**Table 7 foods-11-03540-t007:** Sensory descriptors selected for the assessment of the brandies, their definitions, and the standards used for the training of the panel. The hydroalcoholic mixtures were elaborated using 96% ABV neutral alcohol and demineralized water.

Descriptor	Definition	High Intensity Pattern	Low Intensity Pattern
** *Olfactory evaluation* **			
Aromatic intensity	Intensity of the positive aromatic notes that characterize an aged grape spirit.	P8: VSOP Brandy (4 years), 100% holanda from pot still, hydrated at 30% ABV	P3: 50/50 mixture of P8 and hydroalcoholic mix at 30% ABV
Fruity	Aromas reminiscent of fruits	P9: Pot still holanda hydrated at 30% ABV at a concentration of fatty acid ethyl esters and acetates of higher alcohols above 35 mg/L	P3: 25/75 mixture of P9 and hydroalcoholic mix at 30% ABV
Vanilla	Sweet and delicate note similar to the aroma of vanilla pods, which is caused by the compound vanillin, transferred to the spirit by contact with oak wood.	P9: VSOP Brandy (4 years), 100% pot still holanda, hydrated at 30% ABV and with 10 mg/L vanilla added	P4: Mixture of SCD%65 aged in the three oak species with 5 mg/L of vanilla added, at 30% ABV
Toasted	Characteristic note of toasted wood, reminiscent of baking liquid caramel.	P9: VSOP Brandy (4 years), 100% pot still holanda, hydrated at 30% ABV and with 0.4 g/L caramelized rectified grape must added	P3: A 50/50 mixture of P9 and base brandy from the high pattern without addition, at 30% ABV
Spiced	Olfactory sensation that includes exotic and appreciated oak notes, such as coconut, clove, nutmeg, pepper or cinnamon.	P9: VSOP Brandy (4 years), 100% pot still holanda, hydrated at 30% ABV with the addition of 2 mL/L of hydroalcoholic spiced extract	P3: VSOP Brandy (4 years) hydrated at 30% ABV, with only 1 mL/L spiced extract added
** *Olfactogustatory evaluation* **			
Sweetness	Primary taste most intensely perceived at the tip of the tongue	P8: VSOP Brandy (4 years), 100% pot still holanda, hydrated at 30% ABV and with 3 g/L concentrated rectified grape must added	P3: A mixture of SCD%55 aged in the three oak species, at 30% ABV
Alcohol	Burning sensation in the oral cavity	P9: A mixture of SCD%55 aged in the three oak species at 36% ABV	P4: A mixture of WSD1%55 aged in the three oak species, hydrated at 30% ABV
Smoothness	Warm and velvety sensation in the oral cavity, ending with an easy swallow.	P8: VSOP Brandy (4 years), 100% pot still holanda, hydrated at 30% ABV and with 3 g/L concentrated rectified grape must added	P3: A mixture of SCD%55 aged in the three oak species, at 30% ABV
Oak	Olfactogustatory sensation conferred by oak wood to the brandy and that is characterized by light drying and bitter notes together with a characteristic aroma (retronasal).	P9: SD2%55 aged in *Quercus petraea*, hydrated at 30% ABV	P3: A 50/50 mixture of WSD1%55 aged in *Quercus alba* and *Quercus robur,* at 30% ABV
Balance	Overall assessment of mouthfeel, which defines a structured brandy (full-bodied, with presence), rounded (no outstanding notes, no sharp edges), complex (diversity of notes), with well-integrated alcohol, not remarkable astringency or bitterness, and a long aftertaste.	P9: VSOP Brandy (8 years), 100% pot still holanda, hydrated at 30% ABV	P3: SCD%55 aged in *Quercus alba*, hydrated at 30% ABV

**Table 8 foods-11-03540-t008:** Sensory characterization of the young and aged spirits included in this work. The scores are expressed as the mean *±* standard deviation of the tasting panel.

Sample	Aromatic Intensity	Fruity	Vanilla	Toasted	Spiced	Sweetness	Alcohol	Smoothness	Oak	Balance
WSDD%55	8.0 ± 0.0	8.2 ± 1.2	2.0 ± 1.4	1.3 ± 0.6	1.3 ± 0.5	4.5 ± 0.7	4.0 ± 0.8	7.3 ± 0.6	1.0 ± 0.0	4.3 ± 1.5
WSDD%55QA	7.3 ± 1.2	6.6 ± 1.1	3.5 ± 0.8	3.7 ± 1.2	3.6 ± 1.7	3.5 ± 1.3	4.9 ± 1.2	6.7 ± 1.2	5.3 ± 1.0	5.8 ± 1.3
WSDD%55QP	7.1 ± 1.1	5.6 ± 1.7	3.6 ± 1.0	5.3 ± 0.8	3.6 ± 0.7	2.5 ± 0.6	5.1 ± 0.6	4.5 ± 1.0	7.0 ± 1.1	5.3 ± 1.5
WSDD%55QR	6.1 ± 1.2	5.1 ± 1.4	4.5 ± 1.4	4.0 ± 0.9	3.1 ± 1.0	2.8 ± 1.0	4.3 ± 1.0	6.3 ± 1.2	4.7 ± 1.4	5.3 ± 0.8
WSD1%55	5.0 ± 0.8	6.5 ± 1.9	1.8 ± 0.5	1.3 ± 0.6	1.5 ± 1.0	2.0 ± 0.0	6.0 ± 0.8	4.7 ± 1.5	1.3 ± 0.6	3.3 ± 0.6
WSD1%55QA	6.4 ± 1.2	6.1 ± 1.7	3.9 ± 1.4	3.8 ± 0.8	2.9 ± 1.1	2.8 ± 1.0	5.6 ± 0.5	4.3 ± 0.8	6.5 ± 0.5	4.2 ± 1.0
WSD1%55QP	6.0 ± 0.8	5.1 ± 0.9	4.1 ± 1.6	5.0 ± 1.3	3.4 ± 1.2	2.5 ± 0.6	4.8 ± 1.2	4.2 ± 0.8	6.0 ± 1.3	4.3 ± 1.0
WSD1%55QR	6.0 ± 1.1	4.8 ± 1.0	5.0 ± 0.8	5.0 ± 1.3	2.8 ± 0.5	2.5 ± 0.6	4.3 ± 1.3	5.5 ± 0.8	6.3 ± 1.2	5.2 ± 0.4
WSD2%55	6.0 ± 1.2	6.0 ± 1.4	2.3 ± 1.0	1.7 ± 1.2	1.0 ± 0.0	3.5 ± 0.7	5.0 ± 0.8	5.0 ± 1.0	1.7 ± 1.2	3.0 ± 1.0
WSD2%55QA	5.5 ± 1.3	4.3 ± 1.5	3.9 ± 1.2	4.7 ± 0.5	3.6 ± 1.2	3.5 ± 0.6	4.8 ± 0.7	4.7 ± 1.0	6.0 ± 1.7	4.3 ± 1.0
WSD2%55QP	5.9 ± 1.2	4.8 ± 1.4	4.9 ± 1.7	5.2 ± 1.0	3.6 ± 1.4	2.5 ± 0.6	5.6 ± 1.0	4.7 ± 0.5	5.8 ± 1.0	4.7 ± 0.8
WSD2%55QR	6.8 ± 0.7	4.3 ± 1.4	5.4 ± 1.2	4.5 ± 0.8	3.5 ± 1.3	2.8 ± 1.0	5.4 ± 0.7	4.3 ± 1.2	7.0 ± 0.6	4.2 ± 1.2
SCD%55/SCD%65	5.5 ± 1.1	4.8 ± 1.8	1.8 ± 1.0	1.3 ± 0.6	1.5 ± 0.6	2.5 ± 0.7	7.3 ± 1.0	2.3 ± 0.6	2.0 ± 1.7	2.3 ± 1.2
SCD%55QA	5.8 ± 0.7	4.0 ± 1.6	5.5 ± 1.7	5.0 ± 1.7	3.5 ± 0.8	2.3 ± 0.5	5.5 ± 0.8	3.8 ± 1.2	6.8 ± 1.0	3.8 ± 1.2
SCD%55QP	6.1 ± 1.5	3.4 ± 1.0	4.6 ± 1.9	4.7 ± 0.8	2.7 ± 1.0	2.5 ± 0.6	5.4 ± 1.1	3.8 ± 0.8	6.7 ± 1.2	4.5 ± 1.0
SCD%55QR	6.0 ± 1.9	3.3 ± 1.4	5.0 ± 1.1	4.3 ± 1.2	2.3 ± 1.0	2.3 ± 0.5	5.8 ± 1.3	3.8 ± 1.7	7.0 ± 1.3	3.7 ± 1.4
SCD%65QA	4.3 ± 1.4	4.1 ± 1.1	4.1 ± 1.9	3.8 ± 1.2	3.0 ± 0.8	2.5 ± 0.6	5.5 ± 1.3	3.7 ± 1.0	5.0 ± 1.4	3.8 ± 1.3
SCD%65QP	5.9 ± 1.0	3.4 ± 1.4	4.3 ± 1.8	4.2 ± 1.0	2.6 ± 0.7	2.3 ± 0.5	5.8 ± 0.9	4.2 ± 1.0	7.2 ± 0.8	4.0 ± 1.1
SCD%65QR	4.4 ± 1.2	3.3 ± 1.0	4.0 ± 1.5	3.2 ± 0.8	2.4 ± 0.5	2.0 ± 0.0	5.0 ± 0.8	3.5 ± 1.0	5.7 ± 1.4	3.3 ± 1.0
SD2%55/SD2%65	6.8 ± 0.5	6.7 ± 1.5	1.8 ± 1.0	1.0 ± 0.0	1.3 ± 0.5	3.0 ± 0.0	5.5 ± 1.3	4.7 ± 1.2	2.0 ± 1.7	3.3 ± 0.6
SD2%55QA	5.5 ± 1.3	4.0 ± 1.7	4.5 ± 1.9	4.2 ± 1.2	3.4 ± 1.4	2.5 ± 1.0	5.9 ± 1.2	4.2 ± 1.2	6.8 ± 1.0	4.2 ± 1.2
SD2%55QP	5.8 ± 1.7	3.9 ± 1.0	4.4 ± 1.3	4.7 ± 1.5	3.4 ± 1.3	2.3 ± 0.5	5.6 ± 1.1	3.3 ± 0.8	7.2 ± 0.8	3.7 ± 0.8
SD2%55QR	5.9 ± 1.1	3.8 ± 0.7	5.1 ± 1.8	4.2 ± 1.7	3.4 ± 1.2	2.5 ± 0.6	5.9 ± 0.6	3.8 ± 0.8	7.0 ± 1.3	4.0 ± 0.9
SD2%65QA	5.5 ± 1.3	3.9 ± 1.4	4.8 ± 1.0	4.2 ± 1.0	3.4 ± 1.5	3.0 ± 0.0	5.6 ± 1.2	5.0 ± 1.5	6.5 ± 0.5	4.0 ± 0.6
SD2%65_QP	5.3 ± 1.0	3.5 ± 1.1	4.6 ± 1.3	4.5 ± 1.5	3.9 ± 1.1	2.3 ± 0.5	5.6 ± 1.1	4.5 ± 1.2	5.8 ± 1.5	4.5 ± 1.2
SD2%65_QR	5.4 ± 1.1	4.1 ± 1.5	5.0 ± 1.5	4.8 ± 0.8	3.3 ± 0.7	2.3 ± 1.3	5.9 ± 1.5	3.0 ± 0.9	7.0 ± 0.6	3.7 ± 1.8

**Table 9 foods-11-03540-t009:** Analysis of variance applied to the sensory profile of the brandies. A different letter indicates a significant value difference of the factor for *p* < 0.05 (*). The scores are expressed as the mean *±* standard deviation of the tasting panel.

		Aromatic Intensity	Fruity	Vanilla	Toasted	Spiced	Sweetness	Alcohol	Smoothness	Oak	Balance
**Factor: Distillation system**	*p-*Distillation	0.050 *	0.000 *	0.086	0.763	0.115	0.194	0.002 *	0.000 *	0.005 *	0.000 *
*p-*Distillation x Oak		0.500	0.500	0.729	0.596	0.802	0.793	0.149	0.047 *	0.031 *	0.515
Double pot still distillation (WSDD%55)		6.8 ± 1.6 (b)	5.8 ± 1.8 (c)	3.9 ± 1.5	4.3 ± 1.2	3.5 ± 1.2	2.9 ± 1.0	4.8 ± 1.0 (a)	5.8 ± 1.5 (c)	5.7 ± 1.5 (a)	5.5 ± 1.2 (b)
Simple pot still distillation (WSD1%55)		6.1 ± 1.0 (b)	5.3 ± 1.3 (c)	4.3 ± 1.3	4.6 ± 1.2	3.0 ± 1.0	2.6 ± 0.7	4.8 ± 1.2 (a)	4.7 ± 1.0 (b)	6.3 ± 1.0 (a,b)	4.6 ± 0.9 (a)
Serial distillations with two pot stills. SO_2_ not added (WSD2%55)		6.0 ± 1.5 (a,b)	4.5 ± 1.4 (b)	4.7 ± 1.5	4.8 ± 0.8	3.6 ± 1.3	2.9 ± 0.8	5.2 ± 0.9 (a,b)	4.6 ± 0.9 (b)	6.3 ± 1.2 (a,b)	4.4 ± 1.0 (a)
Continuous distillation in column (SCD%55)		6.0 ± 1.4 (a)	3.5 ± 1.3 (a)	5.0 ± 1.6	4.7 ± 1.2	2.8 ± 1.0	2.3 ± 0.5	5.6 ± 1.0 (b)	3.8 ± 1.2 (a)	6.8 ± 1.1 (b)	4.0 ± 1.2 (a)
Serial distillations with two pot stills. SO_2_ added (SD2%55)		5.7 ± 1.3 (a)	3.9 ± 1.2 (a.b)	4.7 ± 1.6	4.3 ± 1.7	3.4 ± 1.2	2.4 ± 0.7	5.8 ± 1.0 (b)	3.8 ± 0.9 (a)	7.0 ± 1.0 (b)	3.9 ± 0.9 (a)
**Factor: Oak**	*p-*Oa*k*	0.946	0.023 *	0.050 *	0.080	0.273	0.151	0.567	0.024 *	0.722	0.990
*Quercus robur*		6.2 ± 1.4	4.3 ± 1.4 (a)	5.0 ± 1.3 (b)	4.4 ± 1.2	3.0 ± 1.1	2.6 ± 0.7	5.1 ± 1.2	4.8 ± 1.5 (b)	6.4 ± 1.4	4.5 ± 1.1
*Quercus alba*		6.1 ± 1.3	5.0 ± 2.0 (b)	4.3 ± 1.5 (a)	4.3 ± 1.4	3.4 ± 1.2	2.9 ± 1.0	5.3 ± 1.0	4.7 ± 1.4 (b)	6.3 ± 1.2	4.5 ± 1.3
*Quercus petraea*		6.2 ± 1.5	4.5 ± 1.4 (a,b)	4.3 ± 1.7 (a)	5.0 ± 1.1	3.3 ± 1.2	2.5 ± 0.5	5.3 ± 1.0	4.1 ± 0.9 (a)	6.5 ± 1.1	4.5 ± 1.1
**Factor: Aging alcoholic grade**	*p-*Alc. grade	0.007 *	0.947	0.224	0.226	0.982	1.000	0.568	0.534	0.007 *	0.762
*p-*Oak x Distillation		0.228	0.406	0.629	0.440	0.160	0.517	0.647	0.211	0.130	0.728
*p-*Oak x Alc. grade		0.569	0.789	0.767	0.906	0.640	0.348	0.662	0.121	0.571	0.758
*p-*Distillation x Alc. grade		0.137	0.730	0.110	0.086	0.525	0.652	0.925	0.407	0.522	0.480
65% ABV (SCD%65, SD2%65)		5.1 ± 1.3 (a)	3.7 ± 1.4	4.5 ± 1.5	4.1 ± 1.1	3.1 ± 0.8	2.4 ± 0.6	5.6 ± 1.1	4.0 ± 1.3	6.2 ± 1.3 (a)	3.9 ± 1.2
55% ABV (SCD%55, SD2%55)		5.8 ± 1.3 (b)	3.7 ± 1.3	4.9 ± 1.6	4.5 ± 1.5	3.1 ± 0.9	2.4 ± 0.6	5.7 ± 1.0	3.8 ± 1.1	6.9 ± 1.0 (b)	4.0 ± 1.1
**Factor: SO_2_ added in musts**	*p*-SO_2_	0.431	0.050 *	0.926	0.356	0.623	0.059	0.040 *	0.020 *	0.050 *	0.060
*p-*SO_2_ x Oak		0.657	0.727	0.562	0.986	0.991	0.504	0.274	0.471	0.340	0.503
With (SD2%55)		5.7 ± 1.3	3.9 ± 1.2 (a)	4.7 ± 1.6	4.3 ± 1.7	3.4 ± 1.2	2.4 ± 0.7 (a)	5.8 ± 1.0 (b)	3.8 ± 0.9 (a)	7.0 ± 1.0 (b)	3.9 ± 0.9
Without (WSD2%55)		6.0 ± 1.5	4.5 ± 1.4 (b)	4.7 ± 1.5	4.8 ± 0.8	3.6 ± 1.3	2.9 ± 0.8 (b)	5.2 ± 0.9 (a)	4.6 ± 0.9 (b)	6.3 ± 1.2 (a)	4.4 ± 1.0

## Data Availability

Data is contained within the article.

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
