# Peer review of "A Study on the Influence of the Use of Sulphur Dioxide, the Distillation System and the Aging Conditions on the Final Sensory Characteristics of Brandy"

_foods, 2022, doi:10.3390/foods11213540_

Round 1

Reviewer 1 Report

In my opinion, this article is very interesting, the authors analyzed the effect of some variables on the sensory characteristics of brandy. The text is very clear.

Dear authors attached my few suggestions for improvement of the article:

Line 329: In Table 4. Chromatic characteristics and TPI of the young and aged spirits. IPT (mg/L G.A.E.) in place of TPI is reported.

References require standardization, sometimes a DOI number is given, sometimes not, why?

Reviewer 2 Report

I am carefully reviewing the manuscript “A study on the influence of the use of sulphur dioxide, the distillation system and the aging conditions on the final sensory 3 characteristics of brandy”. In this manuscript, many enological factors involving different uses of SO2, distillation system and aging conditions were concerned. They were important but lack of innovation and careful design. This work is more like a technical test rather a scientific research.

Some comments and suggestions are shown below.

1.     Line 69-119. Too much factors concerned in this work (sulphur, the distillation system, the alcohol content during aging, the botanical origin of the aging casks and their toasting degree) But the sample design was too casual. What is the meaning of “light or medium toasted casks” in table 1.

2.     Line 120-137. Why did the authors do the analytical methodologies of TPI, cielab and aroma? How did the authors do the analysis of aroma? I did not find any analysis method about it. But the results did.  If the sensory analysis concerned color, aroma and palate sensation, the qualified chemical analysis methods should be designed here.

3.     Line 155-161. How did the QDA analysis perform? No information can be consulted about training of panel and generation of attributes.

4.     Table 3. The volatile compounds delivered were only alcohols and esters. However, in this work more attention should be paid in volatile phenols and toasted aromas. No compounds of them were analyzed.

5.     If the authors concerned the oak influence, the effect of ellagitannins on bitterness and astringency should analyze. One or two sensory attributes should be focused.
